# Missense variants causing Wiedemann-Steiner syndrome preferentially occur in the KMT2A-CXXC domain and are accurately classified using AlphaFold2

Tinna Reynisdottir[1], Kimberley Jade Anderson[1], Leandros Boukas[2,3]*, Hans Tomas Bjornsson[1,2,4,5]*

1 Laboratory of Translational Medicine, Faculty of Medicine, University of Iceland, Reykjavik, Iceland, 2 McKusick-Nathans Department of Genetic Medicine, Johns Hopkins University School of Medicine, Baltimore, Maryland, United States of America, 3 Department of Biostatistics, Johns Hopkins Bloomberg School of Public Health, Baltimore, Maryland, United States of America, 4 Department of Pediatrics, Johns Hopkins University, Baltimore, Maryland, United States of America, 5 Department of Genetics and Molecular Medicine, Landspitali University Hospital, Reykjavik, Iceland

* lboukas1@jh.edu (LB); htb@hi.is (HTB)

**Data Availability Statement:** Variant data are available in the manuscript and supplementary

## Abstract

Wiedemann-Steiner syndrome (WDSTS) is a neurodevelopmental disorder caused by *de novo* variants in *KMT2A*, which encodes a multi-domain histone methyltransferase. To gain insight into the currently unknown pathogenesis of WDSTS, we examined the spatial distribution of likely WDSTS-causing variants across the 15 different domains of KMT2A. Compared to variants in healthy controls, WDSTS variants exhibit a 61.9-fold overrepresentation within the CXXC domain–which mediates binding to unmethylated CpGs–suggesting a major role for this domain in mediating the phenotype. In contrast, we find no significant overrepresentation within the catalytic SET domain. Corroborating these results, we find that hippocampal neurons from *Kmt2a*-deficient mice demonstrate disrupted histone methylation (H3K4me1 and H3K4me3) preferentially at CpG-rich regions, but this has no systematic impact on gene expression. Motivated by these results, we combine accurate prediction of the CXXC domain structure by AlphaFold2 with prior biological knowledge to develop a classification scheme for missense variants in the CXXC domain. Our classifier achieved 92.6% positive and 92.9% negative predictive value on a hold-out test set. This classification performance enabled us to subsequently perform an *in silico* saturation mutagenesis and classify a total of 445 variants according to their functional effects. Our results yield a novel insight into the mechanistic basis of WDSTS and provide an example of how AlphaFold2 can contribute to the *in silico* characterization of variant effects with very high accuracy, suggesting a paradigm potentially applicable to many other Mendelian disorders.

files. ChIP-seq and RNA-seq are available on GEO GSE99250.

**Funding:** This work was supported by a grant from the Wiedemann-Steiner Foundation to HTB (salary coverage of TR). HTB is also supported by the Louma G. Foundation, the Icelandic Research Fund (#217988, #195835, #206806) and the Icelandic Technology Development Fund (#2010588). Research reported in this publication was supported by the National Institute of General Medical Sciences of the National Institutes of Health (#R01GM121459 to LB). The funders had no role in study design, data collection and analysis, decision to publish, or preparation of the manuscript.

**Competing interests:** I have read the journal's policy and the authors of this manuscript have the following competing interests: HTB is a consultant for Mahzi therapeutics. No other authors have any potential conflict of interest.

## Author summary

Wiedemann-Steiner syndrome (WDSTS) is a neurodevelopmental pediatric disorder caused by the genetic disruption of the histone methyltransferase KMT2A. Since KMT2A has many different domains that perform different functions, we reasoned that by identifying the domains most enriched for WDSTS-causing genetic variants we would gain insights into the incompletely understood molecular pathogenesis of WDSTS. We discovered that the CXXC domain—which binds unmethylated CpGs—shows by far the greatest enrichment, suggesting that loss of the CpG-binding ability of KMT2A plays a central role in WDSTS. Next, to understand specific rules underlying the genetic disruption of the CXXC domain, we combined prior knowledge about the function/structure of the domain with 3D structure prediction by AlphaFold2 to develop an effect classifier for CXXC missense variants. We found that this classifier exhibits accurate performance, and we therefore applied it to provide classifications for any such variant that can possibly arise, in order to aid in the interpretation of such variants in the clinic. Our work provides novel insights into WDSTS and suggests a strategy for missense variant classification that can potentially be applied to many other pediatric genetic disorders.

## Introduction

Wiedemann Steiner syndrome (WDSTS, OMIM: 605130) is a Mendelian disorder of the epigenetic machinery. Its phenotypic features include intellectual disability, postnatal growth deficiency, hypertrichosis, and characteristic facial features. WDSTS is typically caused by heterozygous *de novo* variants in the gene encoding the histone-lysine N-methyltransferase 2A (KMT2A) [1]. *KMT2A*, also known as *MLL/MLL1*, is post-translationally cleaved into N-terminal and a C-terminal fragments, which subsequently heterodimerize. Each of these fragments contains several annotated protein domains. The larger N-terminal fragment contains three AT-hooks, a cysteine-rich CXXC domain, four plant homeodomain (PHD) fingers, a bromodomain and a FYR-N domain. The smaller C-terminal fragment contains a transactivation (TAD) domain, a *Win* motif, FYR-C domain, a SET, and a post-SET domain [2].

In this study, we address two questions. First, how important is the role of the different KMT2A domains in the pathogenesis of WDSTS? The answer would provide important clues into the molecular basis of the disorder, since the different domains mediate different functions. Second, what are the rules that determine how the genetic disruption of the most important domain(s) causes WDSTS? The answer would enable a systematic characterization of WDSTS-causing variants, informing basic biology as well as future clinical decision-making.

To answer the first question, we adopt an unbiased genetic approach. We examine the spatial distribution of likely WDSTS-causing missense variants across the different domains, reasoning that such variants will be enriched in the domains most critical for WDSTS pathogenesis when compared against variants in healthy individuals. To address the second question, we focus on the CXXC domain responsible for binding unmethylated CpGs [3], which we find shows by far the greatest enrichment. By leveraging the recent breakthrough in protein structure prediction from primary amino acid sequence by the deep neural network-based AlphaFold2 [4], we combine accurate 3D structure prediction with existing biological data in order to create an effect classification scheme for missense variants in the CXXC domain. After evaluating our classifier, we deploy it to perform an *in silico* saturation mutagenesis, examining a total of 445 variants.

## Results

### Preferential occurrence of missense variants likely pathogenic for Wiedemann-Steiner syndrome in the CXXC domain of *KMT2A*

If certain domains of a protein are critical for its function, disease-causing missense variants will tend to preferentially occur in these domains. We thus set out to explore the distribution of *KMT2A* missense variants (MVs) across its 15 different domains. We started by identifying 68 MVs that are likely pathogenic for WDSTS (**Methods; Fig 1A**). As a control, we used 1403 MVs which are present in gnomAD and are therefore not expected to cause WDSTS (**Methods; Fig 1A**). We tested each KMT2A domain for enrichment of WDSTS MVs relative to gnomAD MVs. We discovered a 61.9-fold enrichment in the CXXC domain (**Fig 1B**; Fisher's

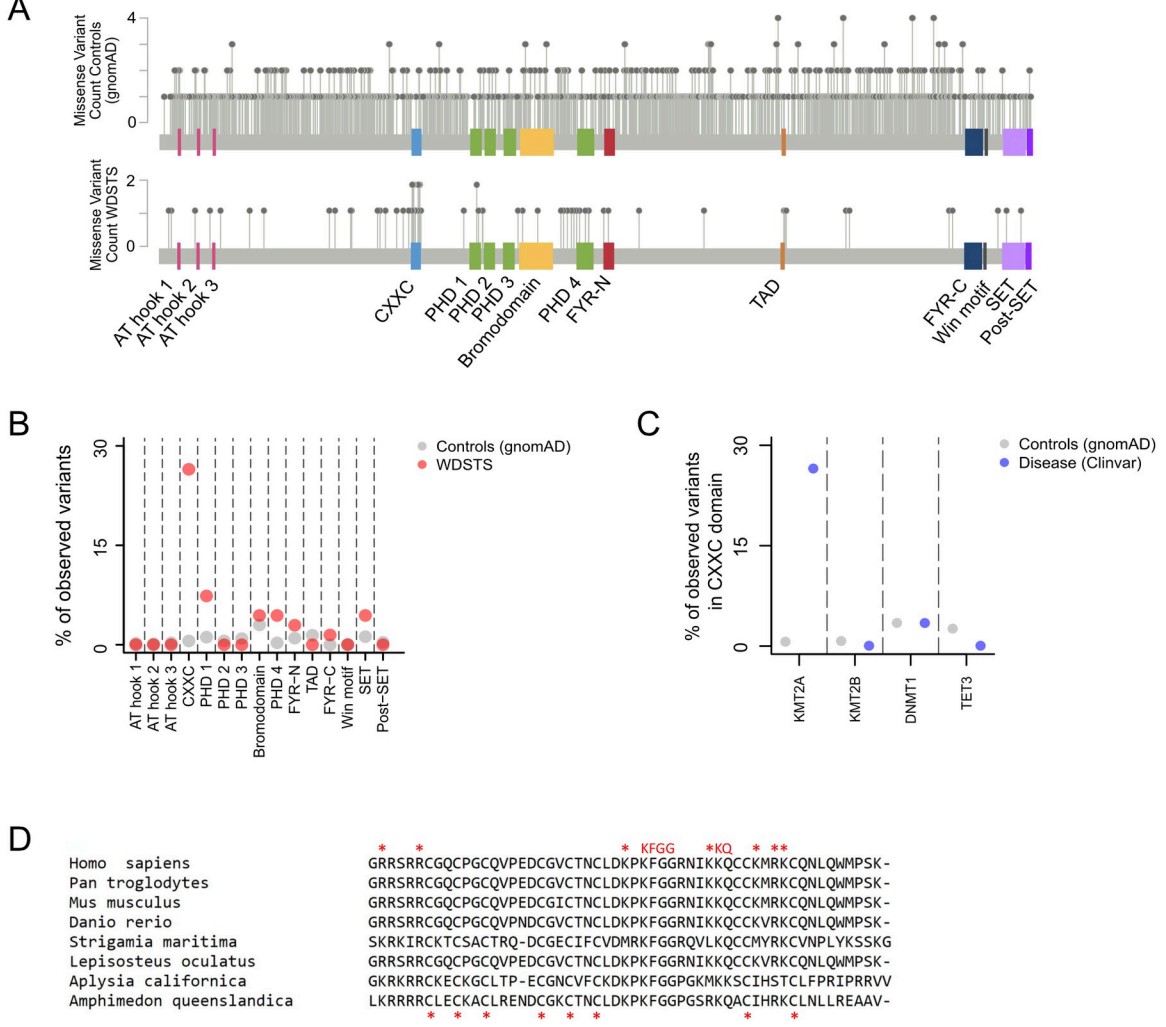

**Fig 1. The distribution of likely pathogenic Wiedemann-Steiner syndrome missense variants across the different domains of *KMT2A*.** (A) *KMT2A* missense variants in gnomAD (top) and WDSTS (bottom). See Methods for filtering criteria. (B) The percentage of missense variants from gnomAD (grey dots) and WDSTS (red dots) that fall in each of the different domains of KMT2A. (C) The percentage of missense variants in gnomAD (grey dots) and likely pathogenic variants (blue dots) that fall in the CXXC domain of different epigenetic regulators. (D) Multiple sequence alignment of the amino-acid sequence of the CXXC domain of KMT2A in eight eukaryotic species. Residues known to be important for DNA binding are marked with red asterisks at the top (see **Methods** for details). The eight zinc ion-binding cysteines are marked with red asterisks at the bottom.

exact test, Bonferroni-adjusted p = 1.56e-18). This far exceeded the enrichment observed at other domains, with the second most enriched domain being the fourth PHD finger (**Fig 1B**; 16-fold enrichment; Fisher's exact test, Bonferroni-adjusted p = 0.043), and the first PHD finger also showing significant enrichment (**Fig 1B**; 6.8-fold enrichment; Fisher's exact test, Bonferroni-adjusted p = 0.031).

To assess the robustness of our result, we repeated our analysis using a control set of 1788 *KMT2A* somatic variants obtained from sequencing of tumor samples (**Methods**). While these variants are likely a mix of driver and passenger variants, the latter are expected to comprise the majority, justifying the use of this set of variants as controls alternative to gnomAD. We recapitulated our result, with the rank ordering of the different domains with respect to their enrichment of WDSTS MVs remaining unchanged (**S1A–S1C Fig**); this is consistent with the pool of somatic variants containing mostly benign passenger variants. However, across all domains, the enrichment estimates are attenuated compared to the gnomAD-vs-WDSTS comparison (**S1C Fig**). This suggests that the same domains that contribute to WDSTS pathogenesis contribute to the tumorigenic role of KMT2A as well. With respect to this tumorigenic role, however, our result should be interpreted with caution; gain-of-function mechanisms that are likely not captured by our analysis are probably also involved, as evidenced from KMT2A translocations that act as drivers in certain types of leukemias.

Notably, the catalytic SET domain of *KMT2A* does not show significant enrichment for WDSTS MVs (**Fig 1B**; Fisher's exact test, p = 0.899). In contrast, it shows significant, albeit weak, enrichment, when the somatic cancer variants are compared against the gnomAD controls (odds ratio = 2.87, Fisher's exact test, p = 8.43e-05). We note here that, while the enrichment of WDSTS-causing MVs within the CXXC domain is consistent with the high conservation of its sequence (**Figs 1D and S2**), conservation alone cannot explain the lack of enrichment in the SET domain, since it is as conserved as the CXXC domain (**S2 Fig**).

## The CXXC domains of other epigenetic regulators do not show enrichment for disease-causing missense variants

We next asked if the preferential occurrence of disease-causing missense variants in the CXXC domain is unique to *KMT2A*, or whether this is a general phenomenon across epigenetic regulators that have this domain. Apart from KMT2A, three other CXXC-domain-containing epigenetic regulators have been linked to Mendelian diseases: KMT2B (DYS28; OMIM:617284), DNMT1 (HSN1E; OMIM:614116), and TET3 (BEFAHRS; OMIM:618798). However, we found that none of these genes exhibits significant enrichment of disease-causing MVs in the CXXC domain (**Fig 1C**; Methods; Fisher's exact test, *DNMT1* p = 1, *KMT2B* p = 1, *TET3* p = 1), and verified that this is not due to inadequate power (**Methods**).

## The disruption of histone methylation in *Kmt2a*-deficient mice preferentially occurs at CpG-rich regions but has no systematic effect on gene expression

Since the CXXC domain mediates binding to clusters of unmethylated CpG dinucleotides [3], our domain enrichment results imply that KMT2A exerts its most important function at CpG-rich regions. Our results also imply that its most important function is not its catalytic activity as a histone methyltransferase, given the lack of MV enrichment in the SET domain. We sought to test these implications, by comparing genome-wide histone methylation (H3K4me1 and H3K4me3) and gene expression patterns in hippocampal CA (Cornu Ammonis) neurons from mice with *Kmt2a* knockout in excitatory neurons (*Kmt2a* cKO) as well as wild-type mice (using ChIP-seq and RNA-seq data; **Methods**) [5].

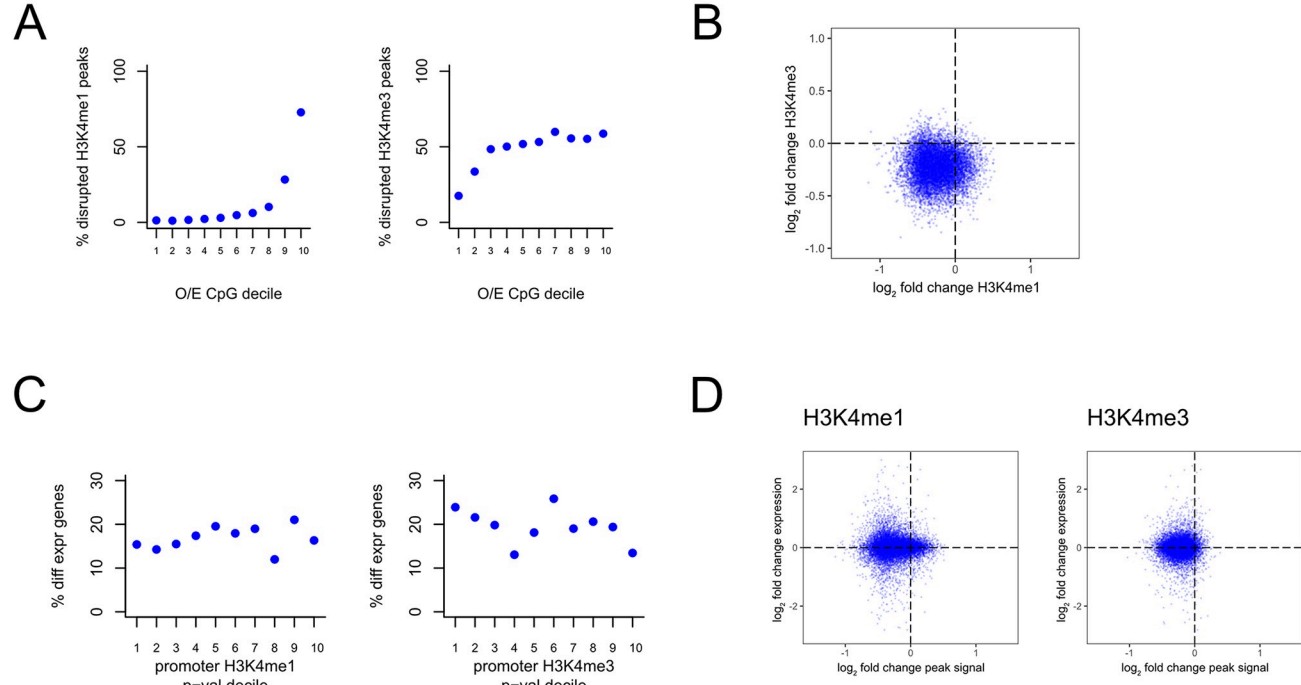

**Fig 2. The relationship between disrupted H3K4me1/3, regional observed-to-expected CpG ratio, and gene expression in *Kmt2a*-deficient mice. (A)** The percentage of disrupted H3K4me1 peaks (left) and H3K4me3 peaks (right), stratified based on the observed-to-expected CpG ratio of the underlying peak sequence. **(B)** Scatterplot of the log$_2$ fold change of H3K4me1 peaks against the log$_2$ fold change of H3K4me3 peaks at promoters (+/-1kb from the TSS) that harbor peaks for both marks. **(C)** The percentage of differentially expressed genes, stratified based on the p-value of associated promoter peaks (+/- 1kb from the TSS) from the differential H3K4me1 analysis (left) and H3K4me3 (right) analysis. **(D)** Scatterplot of the log$_2$ fold change of H3K4me1 peaks (left) and H3K4me3 (right) against the log$_2$ fold change of gene expression of the downstream gene. Each point corresponds to a gene-promoter pair. In cases where multiple peaks were present at the same promoter, the average log$_2$ fold change was computed.

First, we found a strong relationship between the CpG-richness of a region and the probability of H3K4me1/3 disruption. CpG-rich peaks are much more likely to exhibit disruption upon *Kmt2a* cKO compared to CpG-poor peaks (**Fig 2A**; 5 wild-type vs 3 cKO mice; p<2.2e-16; **Methods**). Reassuringly, there is strong concordance between the changes of the two histone marks in the mutant mice; at the vast majority of promoter regions (+/-1kb from TSS) which bear both H3K4me1 and H3K4me3 peaks, both marks show decreased intensity in the mutant mice (**Figs 2B and S3A**). However, although regions with the strongest evidence for histone methylation disruption (1st p-value decile) most frequently correspond to promoters (39% and 86% for H3K4me1 and H3K4me3, respectively; 3.3-fold and 1.3-fold enrichment compared to regions within the 10th p-value decile, respectively; Fisher's exact test, p<2.2e-16; **S3B Fig**), we found no evidence for a systematic impact of promoter H3K4me1 or H3K4me3 disruption on gene expression (5 wild-type vs 6 cKO mice; **Methods**). Specifically, genes whose promoters have disrupted H3K4me1 or H3K4me3 are not significantly more likely to be differentially expressed compared to genes without histone methylation disruption at their promoter (**Fig 2C and 2D**; p = 0.91 and p = 0.13 for H3K4me1 and H3K4me3, respectively, when testing for a shift in p-value distribution between the 1st and 10th deciles with the one tailed Wilcoxon rank-sum test; **Methods**). Taken together, these results show that, at least in adult excitatory CA hippocampal neurons: a) KMT2A preferentially acts at high-CpG-density regions, and b) its catalytic activity has little effect on gene expression, suggesting that it may also not affect higher-level phenotypes.

## An AlphaFold2-based scheme classifies missense variants in the CXXC domain of *KMT2A* with high accuracy

Given our evidence for a central role of the KMT2A-CXXC domain in WDSTS, we sought to develop a variant classification scheme that would: a) label any possible variant in the CXXC domain as pathogenic or not, and b) in the case of pathogenic variants, provide a characterization of their functional effect. We reasoned that such a classifier should take into account the effect of variants on the secondary structures of the domain, the mean Coulombic electrostatic potential of the mutant domain structure, and the ability of the mutated domain to form hydrogen bonds with the DNA backbone.

To assess the feasibility of our approach, we first examined if AlphaFold2 (AF2)—which has recently enabled the determination of 3D-protein structures with experimental-level accuracy—accurately predicts the structure of the CXXC domain. We observed a highly confident prediction (**Fig 3A**; pLDDT>70 for 96.5% of residues and pLDDT>90 for 82.5% of residues;

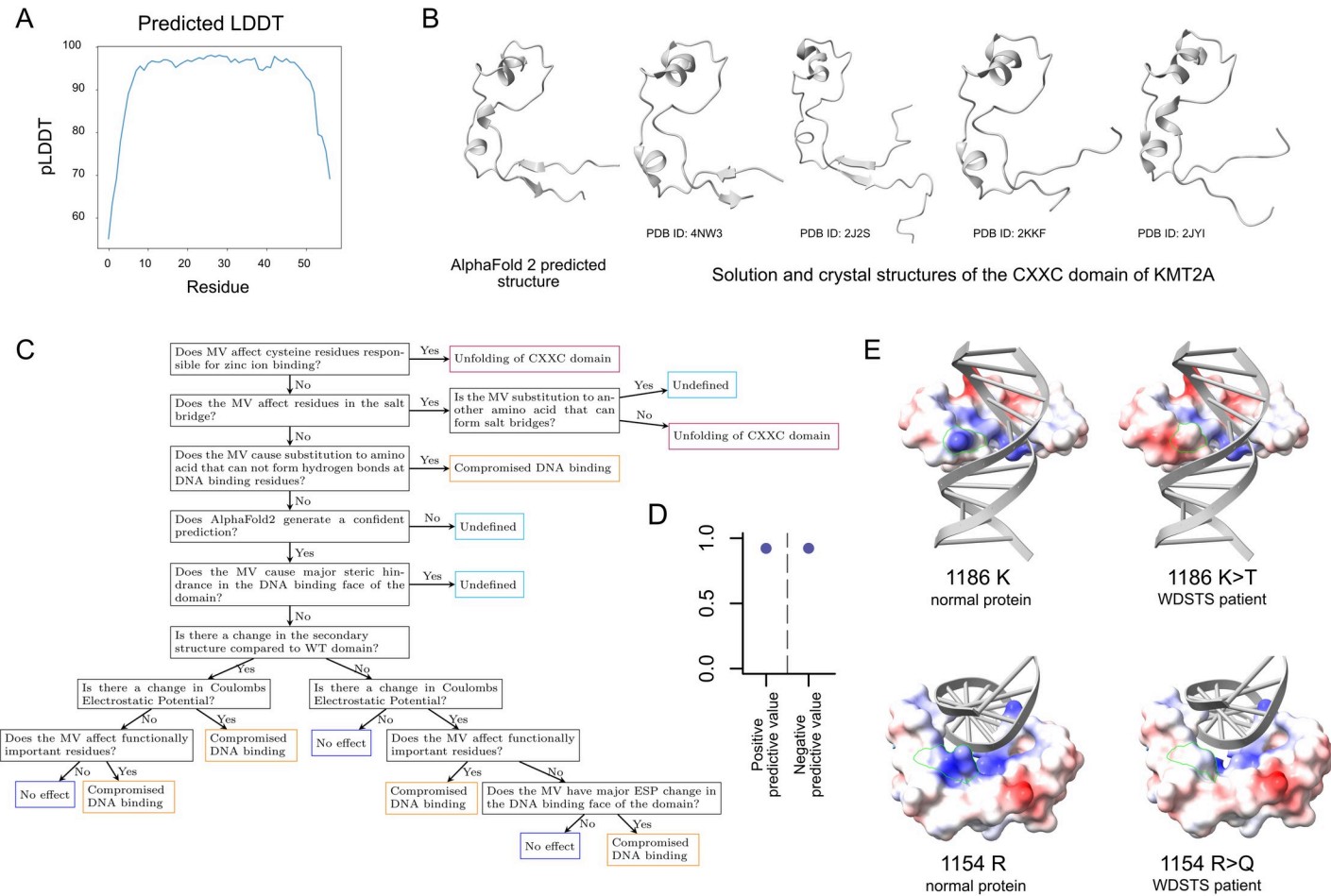

**Fig 3. An AlphaFold2-based variant effect classification scheme for the CXXC domain of KMT2A. (A)** Predicted LDDT values for the CXXC domain of KMT2A. **(B)** The AlphaFold2-predicted and experimentally determined structures of the CXXC domain of KMT2A. **(C)** The variant effect classification scheme. See **Methods** for details on the derivation of the scheme. **(D)** The positive and negative predictive value that the classifier shown in **(C)** attained on a hold-out test set consisting of 41 missense variants (see main text and **Methods** for details). **(E)** The position of two different missense variants in the 3D structure of the CXXC domain, in conjunction with a 3D representation of the engaged DNA backbone. For comparison, the same representation is shown for the normal protein as well (PBD* ID: 4NW3). The surface of the domain is color-coded based on the electrostatic potential, with red indicating a negative charge and blue a positive charge. Both variants lead to a decreased electrostatic potential in a residue important for DNA binding.

**Methods**). In addition, all features previously identified in solution and crystal structures of the domain (ID: 2J2S, 2JYI, 4NW3, 2KKF) are present in the predicted structure: a crescent overall shape, two antiparallel beta sheets at the N and C terminals, and four alpha helices (**Fig 3B**). While no single experimentally derived structure contains all these features, this can be explained by the domain existing in multiple conformational states. We then proceeded to derive a classification scheme, using a training set of 14 MVs with experimentally determined effects and prior biological knowledge about the domain (**Figs 3C and** S6A**, S3 Table; Methods**) [6]. The first decision rules of the scheme pertain to residues whose substitution effects are not captured by AF2 (**Methods**): the eight cysteines responsible for zinc ion binding, the residues that form direct hydrogen bonds with the DNA, and the residues forming the salt bridge. The rest of the scheme is then divided into two cascades, based on the AF2-predicted secondary structure of the mutant domain and its mean Coulombic electrostatic potential (**Methods**). Variants receive different classifications depending on their effect on secondary structure/electrostatic potential, as well as their position.

To evaluate the performance of our classifier, we used a hold-out test set consisting of 41 MVs. Of these, 18 are MVs with strong evidence for pathogenicity for WDSTS (**Methods**), 13 are MVs with experimentally determined effects [6,7], and 10 are MVs seen in gnomAD/TOPMed, and are thus expected to be benign (**S6B Fig, S3 Table**). On this test set, our classifier attained a 92.6% positive and a 92.9% negative predictive value (**Figs 3D, S6B and S6C**). As an example, in **Fig 3E** we depict two WDSTS pathogenic variants which our classifier correctly labels as compromising DNA binding because they decrease the electrostatic potential at residues important for the formation of contacts with the DNA backbone. Notably, out of the 18 WDSTS MVs present in the CXXC domain of *KMT2A*, nine are positioned at cysteine residues responsible for zinc ion binding compared to none of the control variants (Fisher's exact test, p = 0.009816).

### *In silico* saturation mutagenesis classifies 445 variants in the CXXC domain of *KMT2A*

Clinical geneticists seeking to establish a diagnosis for patients are often confronted with missense variants that have not been seen before; this can make it hard to assess if they are pathogenic or not. The accuracy with which our classifier performs motivated us to perform an *in silico* saturation mutagenesis for the CXXC domain of KMT2A. Our goal was to create a resource that will enable rapid classification of any newly encountered missense variant in this domain, so that these classifications can be used as supporting evidence in the clinical setting. We focused on the 50/57 residues that have pLDDT>70 and high MSA coverage. In total, we assessed 450 variants (**Fig 4A**).

Out of these 450 variants, 92 (20.4%) are synonymous, and 27 variants (6%) lead to premature stop codons. The remaining 331 variants (331/450, 73.6%) were classified based on our variant classification scheme. 169 variants (169/331, 51.1%) were predicted to have no effect, 90 (90/331, 27.2%) variants were classified as causing compromised DNA binding, and 67 variants (67/331, 20.2%) were classified as causing unfolding of the domain. For 5 variants (1.5%), we provide no prediction (**Fig 4B**; **S4 Table**). To obtain orthogonal validation for our classifications, we examined the conservation of individual nucleotides coding for the CXXC domain stratified according to the number of substitutions predicted to be damaging, and found strong concordance; sites with a greater number of predicted damaging substitutions are more conserved (**Fig 4C**).

## Discussion

Our contribution in this work is twofold. First, we provide strong genetic evidence that the domains most important for mediating the causal role of KMT2A in Wiedemann-Steiner

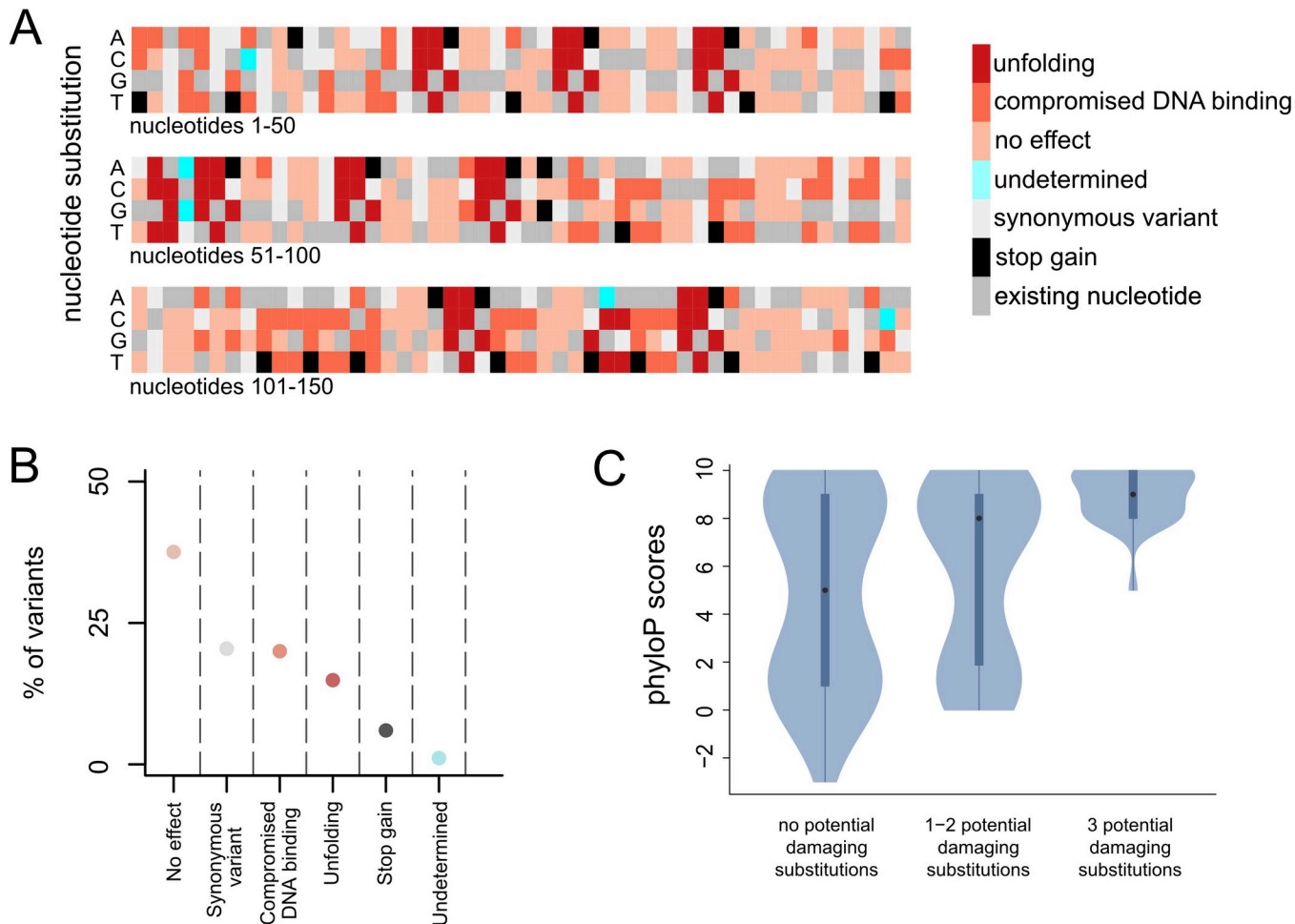

**Fig 4. An *in silico* saturation mutagenesis of the CXXC domain of *KMT2A*. (A)** Heatmap depicting the predicted effect of each nucleotide substitution within the CXXC domain. **(B)** The percentage of variants for each type of predicted effect. **(C)** The distribution of the phyloP score of the nucleotides coding for the CXXC domain, stratified according to the number of substitutions predicted to have a damaging effect (unfolding, compromised DNA binding, stop-gain).

syndrome are the CXXC domain and–to a lesser extent–the fourth and first PHD fingers. It is noteworthy that these PHD fingers have been shown to be important for stabilizing the inter-action between the N- and the C-terminal KMT2A fragments [8]; their enrichment for WDSTS-causing missense variants may thus not be attributable to their histone-binding func-tion. We emphasize here that our domain fold-enrichment estimates are based on a relatively low number of WDSTS variants and may change once more variants are reported. However, we anticipate the rank-ordering of the different domains, and the finding that the CXXC is by far the most enriched domain, to remain unchanged.

Our ChIP- and RNA-seq results suggest that lack of KMT2A recruitment to CpG-rich loca-tions–either by missense variants in the CXXC domain or by loss-of-function variants–is cen-tral to WDSTS pathogenesis. However, they also suggest that the phenotype is not mediated via ensuing systematic defects in histone methylation-dependent gene expression, but rather that alternative mechanisms might be at play. Such mechanisms may involve defects in poly-merase loading, which has previously been shown to depend on the presence, but not the cata-lytic activity, of some epigenetic regulators [9]. Alternatively, KMT2A may serve as a recruiter of other regulatory factors. Our results emphasize that more research aimed at elucidating

such alternative pathways is warranted, and has the potential to contribute to our understanding of WDSTS. This is also consistent with the lack of overrepresentation of WDSTS missense variants within the enzymatic SET domain of KMT2A, and stands in contrast to what was previously observed in Kabuki syndrome, a Mendelian disorder of the epigenetic machinery with considerable phenotypic overlap with WDSTS caused by variants in KMT2D, which is structurally similar to KMT2A [10]. We emphasize here, however, that our findings are based only on data from adult excitatory neurons from the CA region of the hippocampus, whereas gene expression and histone marks are known to be dynamic during development and vary between different cell types. Consequently, future work is needed to assess whether our conclusions generalize to other cell types (neuronal or not) involved in WDSTS pathogenesis, and to earlier developmental stages, when the disease process most likely initiates.

Our second contribution is the demonstration that the recent breakthrough in protein structure prediction by AlphaFold2 can be leveraged in order to classify the effects of missense variants with high accuracy. We highlight that here we use AF2 to directly predict the structure of the mutant proteins; based on these mutant structures, we then assess the effect of variants on secondary structural features and electrostatic potential. This is in contrast to recent work that only uses the wild-type structures as input to algorithms that predict the effect of variants on biophysical attributes like $\Delta\Delta G$ [11]. Prior to our study, it has been unclear if AlphaFold2 is capable of predicting mutant structures accurately, partly because of the use of the multiple sequence alignment. While we do not provide direct evidence, our results indirectly suggest that variant effects on local structural features can be reliably predicted. This conclusion is supported by the high concordance between our predictions and experimentally validated effects, as well as by the severe drop in performance when we use our classifier with inaccurate structure predictions as input (31% negative predictive value; **S7 Fig**) [12]. We do not however, expect AlphaFold2-based classifications to be able to capture global effects of variants, such as destabilization of the entire domain.

There were three variants in our hold-out test set that were classified erroneously. One of them (from the WDSTS cohort) is in fact annotated as a variant of uncertain significance, suggesting that our classification is not necessarily inaccurate in that case, but rather the "true" label may be incorrect. The other two erroneously classified variants may yield insights into possible limitations of our method. The TOPMed variant misclassified as resulting in compromised DNA binding may reflect the inability of our classifier to place effects along a gradient of severity; in other words, this variant may indeed affect DNA binding as we predict, but only to a mild extent not enough to cause severe disease, and is thus still present among healthy individuals. On the other hand, the pathogenic variant misclassified as benign is located in the structurally important KFGG site in the distal loop of the domain. This raises the possibility that our classifier could be improved by including an earlier decision rule assessing whether a variant affects the KFGG site, as we currently only utilize this information in later steps.

Within the context of the ACMG variant interpretation guidelines, our classifier can be used as an *in silico* tool that provides supporting evidence of benign impact (BP4 evidence class) or of pathogenicity (PP3 evidence class). We highlight that it is the first such tool that incorporates accurate 3D protein structure prediction by AlphaFold2, and does not directly use inputs such as conservation and population frequency, which are used by most other tools. Thus, we believe the use of our classifier in combination with other tools may prove particularly powerful for predicting the effect of variants in the CXXC domain of KMT2A.

In summary, our work yields insights into the pathogenesis of Wiedemann-Steiner syndrome and presents a strategy for characterizing variant effects using AlphaFold2 that we anticipate will be broadly applicable to many other disease-relevant proteins.

## Materials and methods

### Missense variants

Missense variants (MVs) present in the general population were obtained from the Genome Aggregation Database (gnomAD; version 2.1.1 and 3.1.1), which does not include individuals with severe pediatric disorders like WDSTS [13]. Somatic MVs were obtained from the Catalogue of Somatic Mutations in Cancer (COSMIC; version 94) [14]. MVs present in individuals with disease phenotypes (Wiedemann-Steiner syndrome [*KMT2A*], Hereditary sensory neuropathy type 1E [*DNMT1*], Childhood-onset dystonia [*KMT2B*], and Beck-Fahrner syndrome [*TET3*]) were obtained from ClinVar [15]. To increase our power, we chose to include ClinVar variants labeled as Pathogenic, Likely Pathogenic, or Variants of Uncertain Significance (VUS), and filtered them using the phred-like CADD scores, acquired from Ensembl Variant Effect Predictor (VEP) [16]. Specifically, we only retained MVs with a phred-like CADD score above 20 [17]. In the case of WDSTS, we obtained 11 additional variants from the Human Gene Mutation Database (HGMD) with CADD score above 20, as well as 16 MVs from Lebrun et al, Baer et al, Miyake et al, WD Jones (all studies of individuals with a WDSTS clinical phenotype; **S1 Table**) [18–22]. Disease MVs that are also present in gnomAD were excluded from subsequent analyses.

To ensure that our domain enrichment estimates are not artifacts driven by the inclusion of VUS's, we also performed the domain enrichment analysis after excluding VUS's and obtained very similar results (**S1D Fig**). Without VUS's, the enrichment in the CXXC domain is even greater, but the confidence interval around the point estimate is wider (**S1D Fig**; confidence interval = 49.2–442.8 without VUS's vs 24.3–173.4 when including VUS's). The same is true for the 1st and 4th PHD finger, which again show significant enrichment as well (**S1D Fig**; For PHD finger 4; confidence interval = 1.9–163.0 without VUS's vs 2.3–97.0 when including VUS's and for PHD finger 1; confidence interval = 1.6–35.0 without VUS's vs 1.9–20.4 when including VUS's). Together, these results are consistent with the notion that, by excluding VUS's, we are increasing our signal-to-noise ratio, since we are not including any variants with uncertain pathogenicity. However, this comes at the expense of less power (reflected in the greater uncertainty around the enrichment point estimates), since this analysis is inevitably conducted using fewer variants.

Finally, since it is possible that very rare MVs in gnomAD may in fact be pathogenic, we assessed whether our result still holds when excluding low-frequency variants, and found this to be true. Using only variants with MAF greater than 10e-5 (452 variants), we found that the CXXC domain still shows the greatest enrichment (odds ratio = 159.4, Fisher's exact test, p = 3.66e-15), followed by PHD fingers 4 and 1 (for PHD finger 4; odds ratio = inf, Fisher's exact test, p = 0.0322 and for PHD finger 1; odds ratio = 5.8, Fisher's exact test, p = 0.122), while the SET domain shows no significant enrichment (Fisher's exact test, p = 0.762), **S1E Fig**.

For the enrichment analysis in the CXXC domain of KMT2B and TET3 (**Fig 1C**), we tested if the observed lack of enrichment can be attributed to the low number of total counts in these genes (22 and 16, respectively). In both cases, we observe 0 MVs in the CXXC domain. In contrast, out of the MVs in gnomAD, 0.7% and 2.5% fall within the CXXC domain of KMT2B and TET3, respectively. Using these gnomAD percentages and the formula for the probability mass function of the binomial distribution, we calculated that, even if the true ratio of the percentage of disease variants falling in the CXXC domain to the corresponding percentage of gnomAD variants is 3 times less compared to the ratio for *KMT2A*, the probability of observing at least one MV in the CXXC domain is 84.5% for *KMT2B* and 99.9% for *TET3*. These estimates make inadequate power an unlikely explanation for the lack of MV enrichment in the

CXXC domain of KMT2B and TET3. The coordinates of protein domains were attained using InterPro [23] (**S5 Table**).

For **Figs 1A** and S1A, variants in *KMT2A* were plotted using the Mutation Mapper tool from cBioPortal [24].

### Evolutionary conservation

The amino acid sequence of the CXXC domain of KMT2A orthologs in the eukaryotic species shown in **Fig 1D** were obtained using the STRING Database [25]. The alignment of the KMT2A orthologous proteins was performed using the Clustal Omega multiple sequence alignment tool from EMBL-EBI [26], with default parameters. The phyloP conservation scores of the SET and CXXC domain nucleotides across 100 vertebrates were obtained using the GenomicScores R package [27].

### ChIP-seq analysis

Raw ChIP-seq sequencing data (fastq files containing unpaired reads) were downloaded from GSE99250 [5]. The reads were aligned to the mouse mm10 (GRCm38) reference genome with Bowtie2 using the default settings with the -U option for aligning unpaired reads, generating a sam file output for each fastq file [28]. The sam files were converted to bam files using Samtools view with the -b option, then sorted using Samtools sort with the -O BAM option for outputting bam files [29]. Peaks were called using MACS2, using a threshold of q-value (-q option) 0.1 for significant peaks, and options "—nomodel—extsize 200" [30]. The sorted bam files and the lists of significant peaks (xls files from MACS2) were used as input to the DiffBind package in R [31]. DiffBind was then run with default settings, except for the options "fragmentSize = 0, RemoveDuplicates = TRUE, filterFun = mean, score = DBA_SCORE_READS" in the dba.count command. Differential peaks obtained in DiffBind were given genomic annotations and converted to GRanges objects using the ChIPseeker package in R [32]. Sequences of peaks were obtained using the getSeq() function from the BSgenome.Mmusculus.UCSC.mm10 package in R [33]. The observed-to-expected CpG ratio was calculated using the following formula [34]:

$$^O\!/_E \; CpG \; ratio = \frac{p(CpG)}{p(C)p(G)}$$

, where $p(CpG)$ represents the proportion of CpGs in a given region (similarly for $p(C)$ and $p(G)$).

### RNA-seq analysis

Raw RNA-seq data (fastq files) were downloaded from GSE99250 [5]. Reads were pseudo-aligned to the mouse mm10 (GRCm38) reference transcriptome using Kallisto [35] (kallisto quant command) with the options—single for single-end reads, -l 150 as an approximation of fragment length, -s 20 as an approximation of fragment length standard deviation, and -b 100 for running 100 bootstraps. The Kallisto output was imported into R using the tximport package [36], with transcripts mapped to genes using the BiomaRt package in R [37]. The tximort software was run with options, type = "Kallisto" and ignoreTxVersion = TRUE. The data was filtered to exclude genes with less than 10 counts. Differential expression analysis was performed using DESeq2 with default settings [38].

To evaluate the robustness and reliability of our differential expression analysis, we performed the following quality control checks. First, we examined the distribution of the

resulting p-values (**S4A Fig**), which is consistent with a two-component mixture; one component corresponding to non-differential genes (p-values distributed uniformly between 0 and 1), and one component corresponding to differential genes (p-values concentrated close to 0). Such a distribution indicates a well-calibrated differential expression test, and the existence of true differential hits. Second, we performed a principal component analysis using the expression matrix after applying a regularized log transformation, as implemented in the rlog() function in DeSeq2 with the setting "blind = TRUE"; this revealed that the mutants are separate from the wild-type on PCA space (**S4B Fig**). Finally, an MA plot (**S4C Fig**) indicates no obvious systematic biases in the expression data.

For **Fig 2A** and **2C**, the percentage of differentially marked H3K4me1/3 peaks and differentially expressed genes were estimated from the corresponding p-value distributions using Storey's method [39,40], as implemented in the qvalue R package. Specifically, we used the pi0est() function, with the "pi0.method" parameter set to "bootstrap".

## Derivation of the variant effect classifier

**AF2 structure predictions.** Structure predictions from AF2 were generated using AF2 Colaboratory (v2.0), which does not use templates (which would be expected to bias the mutant structure towards the structure of the wild-type domain [4]). We do not provide a prediction for variants which result in a structure where: a) the pLDDT value of the beta sheets, positioned at the distal ends, drops below 70, or b) the pLDDT value drops to 85 or lower for more than two residues that are not positioned and the distal ends of the domain, or at residues that fall within a secondary structure that is predicted to be absent.

**Electrostatic potential calculations.** Secondary structure visualization and computation of mean Coulombic values were performed using UCSF ChimeraX (version 1.2.5) [41], with the AF2 predicted structure as input. MVs were classified as causing a change in electrostatic potential (ESP) if the mean electrostatic potential value of the mutant structure deviated by more than 0.2 (in either direction) from the wild-type domain, whose mean value is equal to 6.28.

**Derivation of the decision rules defining the variant effect classification scheme.** We obtained the experimentally determined effects of MVs in the CXXC domain of KMT2A from Allen et al and Cierpicki et al [6,7]. We then used 14 of these variants to form our training set. These variants were chosen to ensure a representation of different types of disruption (unfolding of the domain/defective DNA binding) within the training set. Within a group of variants causing the same type of disruption, a random subset was selected for inclusion in the training set (4 out of 8 variants causing unfolding, 6 out of 11 variants compromising DNA binding), whereas the rest were included in the test set used for evaluating the performance of our classifier (see section below). Based on the training set, as well as existing biological knowledge about the structure and function of the domain [6,7], we derived the decision rules defining our classifier as follows.

Our initial decision rule classifies variants which lead to substitutions of the zinc-ion binding cysteine residues as causing unfolding of the domain, since this was the case for all such variants in the training set (C1161A, C1173A, C1189A, D1166A, R1192A). Our next decision rule then classifies variants which lead to substitutions of amino acids responsible for forming direct hydrogen bonds with the DNA (**S2 Table**; Cierpicki et al [7]) to amino acids that are incapable of forming hydrogen bonds (Ala, Cys, Gly, Ile, Leu, Met, Phe, Pro, Val) as resulting in compromised DNA binding [42]; this was the case with two such variants in the training set (K1186A and Q1187A) and is consistent with the known chemistry underlying protein-DNA contacts.

We subsequently divide the classification scheme into two cascades, based on whether the variant to be classified affects the secondary structure of the domain. For the first cascade, pertaining to variants that do not cause a change in CXXC structure (as determined by Alpha-Fold2; see previous section), we use the change in ESP (computed using ChimeraX; see previous section) and the position of the variant to determine its effect. There were two variants (N1172A, C1188A) in our training set that were experimentally shown to have no effect on the folding of the domain or on DNA binding. We found that these variants caused neither a change in secondary structure, nor a change in ESP. Therefore, we included a decision rule classifying variants that affect neither the secondary structure nor the ESP as benign. There was also one variant (R1153A) that was experimentally shown to have no effect on folding or DNA binding, which we found caused an ESP change without a concomitant secondary structure change, but the ESP change was at a functionally non-important residue (facing away from the DNA backbone and thus unlikely to impact DNA binding). We thus decided to label variants causing no change in secondary structure and changes in ESP at functionally non-important residues as benign. By contrast, there were two variants (K1176A and R1151A) experimentally shown to compromise DNA binding, which we found caused no change in secondary structure, but caused an ESP change at a functionally important residue (that is, a residue that has been experimentally implicated in electrostatic interaction with the DNA backbone, or a residue forming hydrogen bonds with the DNA, or a residue at the structurally important KFGG site (**S2 Table**) [7,43,44]). Therefore, we included a decision rule classifying variants which cause no change in secondary structure but cause ESP changes at these sites as resulting in compromised DNA binding.

Finally, for the second cascade, variants that demonstrate a change in secondary structure as well a change in ESP are classified as compromising DNA binding based on the experimentally determined effect of variants R1154A (loss of beta sheets) and D1175A (loss of an alpha helix). Additionally, similar to the prior cascade, we included a decision rule labeling variants that cause a secondary structure change, do not affect ESP, but affect functionally important residues, as resulting in compromised DNA binding, based on prior knowledge about the function of the domain. Finally, variants that cause secondary structure changes but have no impact on ESP and are not positioned at functionally important residues are classified as having no effect.

**Test set.** As described above, we included a random subset of variants from each potential type of disruption (unfolding/compromised DNA binding). In addition, we included 18 variants seen in patients with phenotypic features of WDSTS [15,18,19,21,22], and 7 and 3 variants from gnomAD and TOPMed [13,45], respectively, which are expected to be benign. With regards to the WDSTS variants, we know that they are pathogenic, but we do not know their precise damaging effect. Therefore, for these variants we are not able to assess whether the precise label that our classifier assigns (unfolding/compromised DNA binding) is true. However, we can still assess the positive predictive value and true positive rate by asking if known damaging variants (pathogenic WDSTS variants or variants with an experimentally determined detrimental effect) are correctly classified as causing some type of damage (either unfolding or disruption of DNA binding).

**Saturation mutagenesis.** In our saturation mutagenesis, we chose to not provide a prediction for the single variant (R1151P) which causes major steric hindrance in the DNA binding face of the domain, since we do not have data to determine the potential impact of this variant. With respect to the MVs that were classified as disrupting the electrostatic potential, the majority caused changes approximately equal to 0.5, with the minimum being 0.31.

## Supporting information

**S1 Fig.** **(A)** *KMT2A* missense variants in COSMIC (top) and ClinVar (bottom). **(B)** The percentage of missense variants in COSMIC (grey dots) and WDSTS patients (red dots) that fall in each of the different domains of KMT2A. **(C)** Correlation between the fold-enrichment (odds ratio) of WDSTS MVs compared to COSMIC MVs and gnomAD MVs. **(D)** The percentage of missense variants in gnomAD (grey dots) and WDSTS patients (red dots) that fall in each of the different domains of KMT2A, after excluding variants of uncertain significance. **(E)** The percentage of missense variants in gnomAD (grey dots) and WDSTS patients (red dots) that fall in each of the different domains of KMT2A, after excluding gnomAD variants with MAF<10e-5.
(PDF)

**S2 Fig. PhyloP score distribution for the SET and CXXC domain nucleotides of KMT2A.**
(PDF)

**S3 Fig.** **(A)** Venn diagram depicting the overlap between promoters (+/- 1kb from TSS) harboring H3K4me1 peaks and those harboring H3K4me3 peaks. **(B)** Genomic annotation of peaks within the 1$^{st}$ (yellow dots) and 10$^{th}$ p-value decile (purple dots) for H3K4me1 peaks (left) and H3K4me3 peaks (right).
(PDF)

**S4 Fig.** **(A)** The histogram of p-values from the differential expression RNA-seq analysis. **(B)** PCA plot based on the expression matrix, after a variance stabilizing transformation (see Methods). **(C)** MA plot of the log$_2$ fold-change against the mean of normalized counts from the differential expression analysis. Differentially expressed genes are colored in blue.
(PDF)

**S5 Fig.** **(A)** Multiple sequence alignment depth plot and **(B)** predicted alignment error of KMT2A CXXC wild-type domain prediction from AlphaFold2.
(PDF)

**S6 Fig.** Variant classification scheme from the **(A)** training set and **(B)** test set. Variants that do not fall under the correct classification according to the scheme are underlined. **(C)** Confusion matrix from test set results.
(PNG)

**S7 Fig. Predicted structure of the CXXC domain of KMT2A using ColabFold without using the multiple sequence alignment or templates as input.**
(PDF)

**S1 Table. WDSTS variant information.**
(CSV)

**S2 Table. Functionally important residues of the CXXC domain of KMT2A.**
(PDF)

**S3 Table. Variant information from training and test set.**
(TXT)

**S4 Table. Saturation mutagenesis results.**
(CSV)

**S5 Table. Amino acid co-ordinates of the proteins domains of KMT2A.**
(PDF)

## Author Contributions

**Conceptualization:** Tinna Reynisdottir, Hans Tomas Bjornsson.

**Data curation:** Tinna Reynisdottir, Kimberley Jade Anderson.

**Formal analysis:** Leandros Boukas.

**Funding acquisition:** Hans Tomas Bjornsson.

**Supervision:** Leandros Boukas, Hans Tomas Bjornsson.

**Visualization:** Tinna Reynisdottir, Kimberley Jade Anderson.

**Writing – original draft:** Tinna Reynisdottir, Leandros Boukas.

**Writing – review & editing:** Leandros Boukas, Hans Tomas Bjornsson.

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
