## [Decision Letter · Decision Letter 0]

15 Mar 2022

Dear Dr Bjornsson,

Thank you very much for submitting your Research Article entitled 'Missense variants causing Wiedemann-Steiner syndrome preferentially occur in the KMT2A-CXXC domain and are accurately classified using AlphaFold2' to PLOS Genetics.

The manuscript was fully evaluated at the editorial level and by independent peer reviewers. The reviewers appreciated the attention to an important problem, but raised some substantial concerns about the current manuscript. Based on the reviews, we will not be able to accept this version of the manuscript, but we would be willing to review a much-revised version. We cannot, of course, promise publication at that time.

If you decide to revise the manuscript for further consideration at PLOS Genetics, please aim to resubmit within the next 60 days, unless it will take extra time to address the concerns of the reviewers, in which case we would appreciate an expected resubmission date by email to plosgenetics@plos.org.

[LINK]

We are sorry that we cannot be more positive about your manuscript at this stage. Please do not hesitate to contact us if you have any concerns or questions.

Yours sincerely,

John M. Greally, D.Med., Ph.D.

Section Editor: Epigenetics

PLOS Genetics

John Greally

Section Editor: Epigenetics

PLOS Genetics

Reviewer's Responses to Questions

**Comments to the Authors:**

Reviewer #1: The work is well designed and it gives evidence from different viewpoints on the relevance of the CXXC domain on the Wiedemann Steiner Syndrome (WSS) pathogenesis. However, it needs some further explanations in order to be published:

Major comments:

- The use of COSMIC variants as evidence of the enrichement of missense variants (MV) should be used cautiously. As authors may know, the translocation of KMT2A, which is part of the pathogenesis of certain leukemias, causes a kind of gain-of-function in KMT2A. It is likely some of these somatic MV may have a similar mechanism, and therefore, it is not applicable to WSS mechanism.

- Albeit ClinVar is a good source of MVs detected in clinical settings, few variants have been extensively curated in terms of interpretation. Considering this, the exclusion of Human Gene Mutation Database (HGMD) as source of confident pathogenic variants must be explained.

- It is seems a bit confusing the inclusion and exclusion of VUSes in the initial analyses. It seems logical to use them when AlphaFold is applied for reinterpretation, not for generating the evidence of the relevance of CXXC domain in WSS.

- Considering recent evidence of the incidence of WSS in newborns (~1/8600 according to Brain. 2020 Apr; 143(4): 1099–1105) and the cutoffs proposed by Whiffin et al. 2017, the MAF cutoff for controls seems too low. Also, it has been demonstrated that genes involved in epigenetic machinery (like KMT2A) are susceptible to clonal hematopoiesis of indeterminate potential (CHIP) and therefore, very-rare control MV may still be pathogenic. These facts are especially crucial to consider, especially when SET domain was found not to be enriched in this work, which may change if another cutoff is considered.

- Although the AlphaFold predicted the best structure of CXXC domain, it would be very graphical and reader-friendly to depict some pathogenic MV detected in silico into the 4nw3 structure, and how they affect the interaction with DNA (which was also determined in that structure).

Minor Comments:

- Please in supp table S4 describe the exact variants following the HGVS nomenclature and their two interpretations (pre- and post- AlphaFold analysis).

- In the Methods section, "Missense variants" subsection, write the genes involved in each of the pehnotypes "(Wiedemann-Steiner syndrome, Hereditary sensory 221 neuropathy type 1E, Childhood-onset dystonia, and Beck-Fahrner syndrome)"

- In Figure 3B, please mention the pdb codes of the different structures.

Reviewer #2: Summary

This study examined variants in KMT2A related to Weidemann-Steiner syndrome to seek insight in to the pathogenesis of the syndrome. First, they identified the locations of pathogenic variants in KMT2A and found them overrepresented in the CXXC domain. This pattern of variant locations was found to be unique to KMT2A, as compared to three other genes which cause Mendelian diseases and did not have pathogenic variants overrepresented in their CXXC domains. To identity the biological impact of KMT2A mutations the authors analyzed previously published data to determine the effect of KMT2A loss on H3K4me3 binding (ChIP-seq) and gene expression (RNA-seq) in WT and KMT2A conditional KO mouse hippocampal neurons. They found that, while H3K4me3 was most disrupted in CpG-rich regions (matching the known function of KMT2A of binding unmethylated CpGs), including at promoters, the expression of nearby genes was not affected by the H3K4me3 loss. Lastly, they used the AlphaFold2 software in conjunction with variants with experimentally verified or in silico predicted pathogenicity to develop an AlphaFold2-based variant classification system with over 90% accuracy, then applied it to most variants in the CXXC domain of KMT2A.

Strengths

• This study has clear formulations of its goals: how important is the role of the different KMT2A domains in WSS pathogenesis and what are rules that determine how the variants – genetic disruption – in these domains causes WSS. To achieve the stated goals it uses simple, quite clear albeit novel methodology in the second goal by linking a genetic change to the protein function through the in-silico modeling of the protein folding.

• By comparing disease causing variant fraction to the control non-pathogenic variant fraction in the KMT2A domains the study identifies the CXXC domain as having the most significant role among other domains in WSS pathogenesis.

• Using KMT2A-deficient mice to show that H3K4me1 disturbances preferentially occur at CpG-rich regions, but has no systematic effect on gene expression (via ChIP and RNA-seq)

Major Weaknesses

• It is recommended to characterize the conservation properties of the domain across species in the orthologous genes. That would provide stronger evidence of the importance of CXXC domain and its intolerance to perturbations. This is missing in the study as well as the genomic coordinates and identities of the transcripts (or a single major transcript) for which the missense variants were collected.

• In the methods part the validation of the proposed classification scheme is poorly described. How exactly the training of the rule based system was performed using the 14 variants for training to obtain the rules shown in Figure 3 C ? How many splits of the data were performed in hold-out cross-validation? Just one or more? The authors should provide a crosstabulation – a confusion matrix- for validation and for the training data along with the accuracy estimates in Figure 3 D. That would help to understand the class balance in the subset of variants used for derivation of the rules.

• What is the rationale of the In silico saturation mutagenesis of 445 variants? It is not clear whether these variants were created artificially and then classified or they were taken from the existing resources. What is the rational of this analysis, how it is useful for KMT2A variant interpretation?

• The ChIP-seq and RNA-seq data were downloaded from paper that uses a mouse model that deletes KMT2A conditionally in adult excitatory forebrain neurons and hippocampal CA. Therefore, it is hard to make firm conclusions around the impact of loss of KMT2A on gene expression in WDSTS as it is a neurodevelopmental disorder and therefore KMT2A may have different targets and effects on chromatin and gene expression during development rather than in the adult brain that has already developed. Conclusions around ChIP-seq and RNA-seq data regarding deposition of histone methylation and gene expression are generalized; authors should address temporal limitation of their mouse model. Additionally, the mouse model used is only KO in neurons. This should also be considered as a limitation as gene expression is different in different brain cell types and loss of KMT2A in these other cell types may also contribute to pathology (either cell autonomously or non- autonomously, meaning loss in other cells impacts neurons as well)

• The source of the ChIP-seq and RNA-seq data (reference 4) includes ChIP-seq for H3K4me3 along with H3K4me1. Is there a particular reason only the H3K4me1 data was used?

• The author’s focused their RNA-seq analysis on genes in relation to H3K4me1 and find no systematic impact of H3K4me1 disruption on gene expression. As a positive control for the analytical approach used in this study, authors should confirm that their analytical pipeline can first replicate (within reason, expecting some variation due to differences in analytical pipelines) the previous published data (reference 4) which found “471 genes down- and 225 genes upregulated in Kmt2a cKO”?

Minor Weaknesses / Corrections

• Not clear what Figure 4 represents. It would be clearer if the domain CXXC would be represented as a track in the genome browser with overlaid variant classifications.

• It would helpful if the numerical summaries of the features used in the derivation of the classification scheme would be provided along with the training and testing data.

• It would be helpful if the proposed scheme would be discussed in view of ACMG variant interpretation guidelines. Which evidence class this method could potentially support?

• Suggest the addition of a sentence describing AlphFold2 in the introduction or a reference to guide the reader.

• The OMIM abbreviation of the syndrome is WDSTS

• From the hold-out test set on the classifier, can you elaborate on the variants that were classified incorrectly? Is there rationale to explain the misclassifications or can you comment on how these misclassified variants may highlight a limitation of this classification scheme?

• The authors should expand on their discussion at lines 187-191 to explain how their results suggest that lack of KMT2A is both “central to WSS pathogenesis” but also that “the phenotype is not mediated via ensuing defects in the deposition of the histone methylation”, in the context of what “alternative mechanisms might be at play”.

• The first reference for AlphaFold2 (ref 36) is given at line 289 in the Methods but should be given earlier such as at line 135 in the Results.

• Supplemental Table 4, add 2 columns to indicate the nucleotide position and amino acid position for each row/variant. In the text or table legend specify the transcript and protein accession numbers used for the analysis.

• The authors mentioned positive and negative prediction values for the classifier. It is confusing and can be misleading as the term “prediction value” is also often used to literally mean the predicted values, like in a regression model for instance. I assume they meant these are true positive and true negative rates (sensitivity and specificity) based on the classifier test results. It would better if they can clarify this.

**Have all data underlying the figures and results presented in the manuscript been provided?**

Reviewer #1: **No: **There is no full list of variants tested in silico, which would be really useful for diagnostic laboratories.

Reviewer #2: Yes

PLOS authors have the option to publish the peer review history of their article (what does this mean?). If published, this will include your full peer review and any attached files.

Reviewer #1: **Yes: **Victor Faundes

Reviewer #2: No

---

## [Decision Letter · Decision Letter 1]

27 May 2022

Dear Dr Bjornsson,

We are pleased to inform you that your manuscript entitled "Missense variants causing Wiedemann-Steiner syndrome preferentially occur in the KMT2A-CXXC domain and are accurately classified using AlphaFold2" has been editorially accepted for publication in PLOS Genetics. Congratulations!

Yours sincerely,

John M. Greally, D.Med., Ph.D.

Section Editor: Epigenetics

PLOS Genetics

John Greally

Section Editor: Epigenetics

PLOS Genetics

Comments from the reviewers (if applicable):

Reviewer's Responses to Questions

**Comments to the Authors:**

Reviewer #1: In my opinion, the issues raised were addressed accordingly.

**Have all data underlying the figures and results presented in the manuscript been provided?**

Reviewer #1: Yes

PLOS authors have the option to publish the peer review history of their article (what does this mean?). If published, this will include your full peer review and any attached files.

Reviewer #1: **Yes: **Victor Faundes

**Data Deposition**

http://datadryad.org/submit?journalID=pgenetics&manu=PGENETICS-D-22-00064R1

**Press Queries**

---

## [Editor Report · Acceptance letter]

16 Jun 2022

PGENETICS-D-22-00064R1 

Missense variants causing Wiedemann-Steiner syndrome preferentially occur in the KMT2A-CXXC domain and are accurately classified using AlphaFold2 

Dear Dr Bjornsson, 

We are pleased to inform you that your manuscript entitled "Missense variants causing Wiedemann-Steiner syndrome preferentially occur in the KMT2A-CXXC domain and are accurately classified using AlphaFold2" has been formally accepted for publication in PLOS Genetics! Your manuscript is now with our production department and you will be notified of the publication date in due course.

With kind regards,

Agnes Pap

PLOS Genetics

On behalf of:
